# Prevalence of Essential Nutrient Supplement Use and Assessment of the Knowledge and Attitudes of Lebanese Mothers towards Dietary Supplement Practices in Maternal, Infancy and Preschool Ages: Findings of a National Representative Cross-Sectional Study

**DOI:** 10.3390/foods11193005

**Published:** 2022-09-27

**Authors:** Hala Mohsen, Carla Ibrahim, Khlood Bookari, Danielle Saadeh, Ayoub Al-Jawaldeh, Yonna Sacre, Lara Hanna-Wakim, Marwa Al-Jaafari, Marwa Atwi, Sabine AlAsmar, Jessica Najem, Maha Hoteit

**Affiliations:** 1Faculty of Public Health, Lebanese University, Beirut P.O. Box 6573, Lebanon; 2Doctoral School of Sciences and Technology (DSST), Lebanese University, Hadath P.O. Box 6573, Lebanon; 3PHENOL Research Group (Public HEalth Nutrition Program Lebanon), Faculty of Public Health, Lebanese University, Beirut P.O. Box 6573, Lebanon; 4Lebanese University Nutrition Surveillance Center (LUNSC), Lebanese Food Drugs and Chemical Administrations, Lebanese University, Beirut P.O. Box 6573, Lebanon; 5Department of Nutrition and Food Sciences, Faculty of Arts and Sciences, Holy Spirit University of Kaslik (USEK), Jounieh P.O. Box 446, Lebanon; 6Department of Clinical Nutrition, Faculty of Applied Medical Sciences, Taibah University, Madinah 42353, Saudi Arabia; 7National Nutrition Committee, Saudi Food and Drug Authority, Riyadh 11451, Saudi Arabia; 8INSPECT-LB (Institut National de Santé Publique d’Épidémiologie Clinique et de Toxicologie-Liban), Beirut P.O. Box 1103, Lebanon; 9World Health Organization Regional Office for the Eastern Mediterranean, Cairo 11371, Egypt; 10Department of Agricultural and Food Engineering, School of Engineering, Holy Spirit University of Kaslik (USEK), Jounieh P.O. Box 446, Lebanon; 11University Medical Center, Lebanese University, Beirut P.O. Box 6573, Lebanon

**Keywords:** knowledge, attitudes, practices, dietary supplements, mothers, under-5 children, Lebanon

## Abstract

Mothers are understandably concerned about protecting the well-being of their offspring in every way possible, including providing oral dietary supplements (DSs). Up to now, there has been limited data on maternal knowledge and attitudes toward concomitant maternal–child DSs practices in Lebanon. This study evaluated the maternal knowledge and attitudes toward DSs and documented the DS-related practices in mothers and their under-5 children with their correlates. This cross-sectional study involved a representative stratified cluster random sample of 511 mother–child dyads (mothers: mean age ± SD = 30.25 ± 4.98 years; children: mean age ± SD = 18.7 ± 15.5 months, girls: 55.0%). A self-administered questionnaire was used to meet study aims. Most mothers lack awareness and hold unfavorable attitudes regarding DS use. Among all mothers, 47% were DS users, with the majority using vitamin D (82%). Almost 64% of mothers provide DSs for their children, with a predominant use of multivitamin drops (61.0%). “To keep the child healthy” was the reported reason by 60.0% of mothers to provide DSs for their children. Physicians were the primary information source about DSs for most mothers (64.0%). The usage of DSs among mothers was influenced by their pregnancy status, child’s age, number of children per household, and their awareness and attitudes towards DSs. DS usage among children was correlated with maternal DS use and their mothers’ attitudes towards DSs. DS usage among Lebanese mother–child dyads is common. Mothers should be the focus of education sessions regarding DS use.

## 1. Introduction

Around three-quarters of American adults are taking dietary supplements (DSs) each year [1]. Similarly, 70% of Lebanese adults reported consuming DSs while 51% of DS users were females [2]. According to “the C.S. Mott Children’s Hospital National Poll on Children’s health” report [3], parents who take DSs are more likely to have children doing so. For instance, 1 in 2 parents say that their child takes DSs regularly, 80% of them say they chose products “made specifically for children”, and 43% say they referred to their child’s healthcare provider before using DSs [3]. The main reason behind providing DSs for their offspring was the parent’s feeling that their child does not eat fruits, vegetables, and a well-balanced diet [3].

During infancy and early childhood, nutritional deficiencies could cause adverse health effects with long-term impacts, including rickets, iron deficiency anemia, goiter, obesity, coronary heart disease, type 2 diabetes, stroke, cancer, and osteoporosis [4]. Thus, according to the American academy of pediatrics, when dietary needs could not be met through food consumption, one alternative approach to addressing nutrient inadequacies among infants and children is to provide oral DSs [5]. DSs are defined as non-drug products intended to supplement the diet with vitamins, minerals, herbs, and amino acids, according to the United States Food and Drug Administration (U.S. FDA) [6]. The scientific consensus is that DSs are formulated to mitigate nutrients shortfalls rather than replace the diet, and they are not meant to treat, identify, stop, or cure illnesses [7]. On the other hand, certain health risks are unique to children who use DSs, including incorrect dosing (depending on the child’s age and body weight), side effects, drug–dietary supplement interactions, and severe allergic reactions [8]. The prevalence of DS use among infants and children in the United States was 33% and the most commonly used type of DS was vitamins A, C, and D, calcium, and iron [9]. Given that knowledge deficits on oral dietary supplementation in the adult Lebanese population is a typical occurrence as evidenced by preliminary research efforts [2], Lebanese mothers are supposed to be at risk of providing unnecessary supplement products for their children and even in toxic doses due to misconceptions and lack of understanding. As mothers are the primary caregivers in most families, their level of awareness of the oral dietary supplementation topic affects the usage of DSs in their children. Until recently, no studies have characterized the maternal knowledge and attitudes towards oral DS practices at maternal, infancy, and preschool ages. Not only that, as far as we know, the data on the use of DSs among mothers and the pediatric population is currently scant worldwide, even not available nationally and in the Middle East and North Africa (MENA) boundaries. Therefore, we are providing the findings of this research study to be the first to (1) evaluate the maternal knowledge and attitudes toward oral dietary supplementation; (2) document the DS-related practices in mothers and their under-5 children; and (3) identify the correlates influencing the intake of DSs in both mothers and their under-5 children, in Lebanon.

## 2. Materials and Methods

### 2.1. Study Design, Sampling Procedure, and Eligibility of Participants

The present investigation is a cross-sectional study conducted between January and December 2021. Using a stratified cluster sampling technique, we reached a representative stratified random sample of children aged 5 years and younger of both genders (*n =* 511) with their mothers, collected from Lebanese households. Study participants (mother–child dyads) were purposely recruited from the main eight Lebanese provinces: Beirut, Mount Lebanon, North Lebanon, Akkar, South Lebanon, Nabatieh, Beqaa, and Baalbeck-Hermel. Accordingly, the strata were the mentioned Lebanese governorates, and clusters were recruited at the district level. The number of households recruited was determined based on the district’s population density, referring to the probability proportional to size technique, with more participants from the more crowded districts. Because of the national COVID-19 pandemic lockdown imposed at the time of data collection, face-to-face interviews were not possible, and eligible mothers were asked to fill out an online form of a self-administered questionnaire. Announcements of the study were performed using different social media platforms (WhatsApp, Facebook, Instagram, and Twitter) to invite mothers to participate. To be included in the study, mothers had to be: (1) Lebanese; (2) aged between 18 and 49 years old; and (3) had at least one child aged 5 years old and younger. Otherwise, mothers were excluded, and their data were not considered for analysis.

### 2.2. Sample Size Calculation

This study was conducted to meet a myriad of objectives apart from oral dietary supplementation assessment among mother–under-5 child dyads, including the examination of the feeding patterns and the evaluation of the nutrition status of the mothers and their children. The findings on the feeding patterns and malnutrition assessment in this sampled population are published elsewhere [10].

According to the Lebanese Ministry of Public Health (MOPH) latest population estimates [11], the total number of mothers of childbearing age across all Lebanese governorates in 2019–2020 was 1,551,344. As per the Epi-info statistical software developed by the Center for Disease Control and Prevention [12], taking an acceptable margin of error of 5%, statistical power of 80%, and a 40% expected frequency of infant formula consumption among children younger than 24 months of age in LMICs [13], the minimum representative sample size consisted of 369 mothers, to ensure appropriate power for statistical analyses. Subsequently, we eventually increased this number by about 50%, taking into account the cluster effect and the refusals to participate, reaching 511 mother–child dyads to be included in our study.

### 2.3. Data Collection: Assessment of Oral Dietary Supplementation

This study employed a validated questionnaire retrieved from the Center for Diseases Control (CDC) website [14] and adapted to our sampled population, taking into account cultural considerations. The questionnaire was translated by experts in the field into Arabic, the native language of the participants, and then tested in a focus group to ensure that the original meaning of the questionnaire was retained. In the current study, the oral dietary supplementation-related data and questionnaire parts that align our study objectives were presented. To meet the current study objectives, information from mothers was collected on the following characteristics: (1) mother’s age, mother’s self-reported body weight, and height, nationality, current residence, marital status, number of children in the household, whether the mother has twins or triplets, current pregnancy status, monthly family income, mother’s employment status and work discipline (health field/non-health field), mother’s COVID-19 infection status (currently infected or before the data collection); (2) child’s age, gender, birth weight, birth height, and child’s COVID-19 infection status (currently infected or before the data collection), the child’s status in terms of ever breastfeeding, exclusive breastfeeding, and complementary feeding; and (3) total number of household members excluding newborns and the total number of rooms excluding the kitchen and the bathroom to calculate the household’s crowding index. As for the oral dietary supplementation section, collected data included the following details: (1) mother’s knowledge of oral dietary supplementation; (2) mother’s attitudes towards multiple aspects related to oral dietary supplementation (options range from strongly agree to disagree strongly; re-categorized in two response options (agree/disagree) to facilitate data presentation and analysis); (3) mother’s oral dietary supplementation-related practices (including the use of DSs currently or previously); (4) child’s oral dietary supplementation-related practices (including the use of DSs currently or previously); (4) frequency and duration of use of different oral DSs among mothers and their children; (5) types of supplements used by mothers and their children; (6) reasons that led mothers to supplement their children’s diet; and (7) adverse effects experienced by mothers as a result of oral dietary supplementation. Before conducting the survey, a focus group (*n* = 20) meeting was enrolled to evaluate the feasibility of each question.

### 2.4. Ethical Considerations

The study was carried out in accordance with the Helsinki Declaration’s ethical guidelines. The Al-Zahraa University Medical Center’s Ethics Committee, reference Nb 9-2020, in Beirut, Lebanon, provided their ethical approval to conduct this study (issued date: 2 December 2020). Mothers were asked to sign a written consent form outlining the study’s objectives, their participation rights, including privacy and confidentiality concerns. The participation was voluntary, with no obligation to do so, and mothers had the right to withdraw at any time of the study. No identification information was collected from mothers, and their responses were anonymous. The estimated time for completion of the survey was 15–20 min. The study’s research team was available to respond to any inquiries via the WhatsApp, Messenger, and Zoom applications.

### 2.5. Statistical Analyses

We performed the statistical analysis of this study using the Statistical Package of Social Sciences Software (SPSS) (Version 25.0. IBM Corp: Armonk, NY, USA). A “weighting” variable was created to adjust the representation of the sampled population according to all the governorates. Respondents’ characteristics were presented as frequencies (percentages) for categorical variables, while means ± standard deviation (SD) for continuous variables. The chi-squared test (χ2) was used to determine the associations between study variables. Further, the backward stepwise regression analytical method was used to investigate the extent of association of the most significant variables with the DS usage among mothers and their under-5 children in this study. A *p*-value of less than 0.05 was considered significant for all analyses.

## 3. Results

### 3.1. General Characteristics of Mothers and Their Under-5 Children in This Study

From 541 recruited mother–child dyads, 511 were eligible to participate (response rate: 94.4%). The mean age ± SD of mother participants was 30.25 ± 4.98 (years), with the majority (88.6%) being adults aged between 25 and 49 years old. The mean Body Mass Index (BMI) ± SD of mothers was 24.85 ± 4.53 kg/m^2^. Around half of them (52.0%) had a normal body weight, whereas 43.8% were overweight or obese. Only a few (4.2%) were underweight. More than half of the mothers (56.4%) were residents in the Beirut and Mount Lebanon districts. Almost all of them (99.6%) were married, and the remaining (0.4%) were divorced. Further, 47.2% had only one child, 46.6% had 2–3 children, and 6.2% were mothers of 3 children and more. Only 5% were mothers of twins or triplets. Among all mothers, 10.3% were pregnant at the time of the data collection. More than half (59.6%) reported a monthly family income ranging between LBP 750,000 and 2,250,000. Nearly half of the mothers (43%) were currently working, of which 25.6% were healthcare workers. Regarding COVID-19 history, about one-quarter of mothers (24.1%) have had the infection. The household crowding index (mean ± SD) was 1.03 ± 0.40 (Table 1).

As for the child participants, the overall mean age ± SD was 18.7 ± 15.5 (months), with boys (mean age ± SD:18.0 ± 15.5 months) being younger than girls (mean age ± SD:19.5 ± 15.6 months), *p =* 0.31. Most children (36.4%) were aged between 1 and 3 years old, 25.3% between 0 and 6 months, 21.3% between 6 and 12 months, and 17.0% between 3 and 5 years old. The overall mean weight and height ± SD at birth were 3168.9 ± 617.9 g and 49.5 ± 5.2 cm, respectively. However, girls had a higher birth weight than boys (3283. 35 ± 594.2 g vs. 3074.18 ± 622.2 g, *p <* 0.001). As for the COVID-19 history of the children, 16.4% have been diagnosed with the disease. Among all, 95.2% of the under-5 children were ever breastfed, 59.1% had been breastfed exclusively in the first 6 months of the child’s life, and 48.2% had their complementary feeding at 6 months of the child’s age or older (Table 2).

### 3.2. Mothers’ Knowledge and Attitudes towards Oral Dietary Supplementation

Mothers responded to multiple questions on their knowledge and attitudes towards oral dietary supplementation. Among all, 57% of mothers did not perceive the sufficiency of their food nutrients’ intake in meeting the daily dietary requirements and around 75% of mothers denied the statement that herbal products (HPs) can cause adverse effects when consumed with drugs. Furthermore, most mothers (68%) consented that HPs are safe to be consumed because they are from natural sources. Around 50–67% of the mothers disapproved that the effectiveness and safety of DSs were tested through clinical trials then sold at pharmacies. Most mothers believed that DSs are necessary to maintain good health (62%) and protect against infections (58.5%). Most mothers disbelieved that DS consumption could cause adverse effects and that the consumption of DSs might mitigate the negative side effects of smoking, alcohol consumption, and being physically inactive (75.0% and 86.6%, respectively) (Figure 1a,b).

### 3.3. Practices Related to Oral Dietary Supplementation among Mothers

When asked about their use of DSs, about half of the mothers (47%) reported using one or more supplement products. Among supplement users, most mothers reported using vitamin D (82%), followed by iron (67%), calcium (61%), multi-vitamin-mineral (56%), magnesium (47%), vitamin C (41%), vitamin B12 (31%), zinc (30%), vitamin B6 (pyridoxine) (30%), and folic acid (29%) supplements. Others reported the use of vitamin E (19%), vitamin B1 (thiamin) (19%), vitamin A (18%), vitamin B2 (riboflavin) (16%), iodine (16%), niacin (15%), phosphorus (15%), choline (13%), selenium (13%), and antioxidant (12%) supplements (Figure 2).

In terms of other DS-related practices, 52% of mothers admitted to looking on the DSs’ label most of the time before using it, others reported doing so occasionally (23.1%) and rarely (18.1%), respectively, while 6.6% reported never looking at the supplement’s label. The majority of mothers (84.2%) supplement their diets because of physician prescription and purchase supplement products from pharmacies (95.9%). When asked to estimate the money spent per month on purchasing DSs, 26.7% of mothers reported spending less than LBP 50,000 (equivalent to less than USD 33.5), 48.9% between LBP 50,000 and 100,000 (equivalent to USD 33.5–66.5), 18.8% between LBP 100,000 and 200,000 (USD 66.5–133.5), and 5.4% more than LBP 200,000 (or more than USD 133.5) (Appendix A).

### 3.4. The Frequency and Duration of Use of Single Vitamin/or Mineral and Multi-Vitamin-Mineral Supplements among Mothers

Table 3 reveals that iron supplements were the supplement products that mothers used regularly on a daily basis (50.4%), followed by calcium (39.3%), multi-vitamin-mineral (MVM) (39.2%), vitamin D (30.2%), magnesium (29.4%), vitamin C (20.3%), folic acid (19.4%), vitamin B6 (16.1%), vitamin B12 (15.3%), zinc (14.5%), vitamin E (12.0%), vitamin A (11.6%), vitamin B1 (10.7%), vitamin B2 (9.4%), niacin (9.1%), iodine (9.1%), selenium (8.3%), phosphorus (7.9%), choline (7.0%), and antioxidant (6.2%) supplements. Moreover, vitamin D supplements were the most used weekly, 2–3 times per week (27.3%) or once per week (14.5%). Only a few mothers use any supplement products monthly, except for vitamin D supplements which were reported to be used by almost 10% of mothers once per month.

### 3.5. Practices Related to Oral Dietary Supplementation among under-5 Children

Regarding DS use among under-5 children, 64% of mothers reported supplementing their child’s diet. Among children using DSs, multivitamin drops were the most commonly provided for 61%, followed by iron (60%), vitamin C (28%), vitamin A (23%), and calcium (22%) supplements. Other DSs were also provided for the under-5 children including vitamin E (17%), vitamins B1 (thiamin) (16%), B2 (riboflavin) (16%), B6 (15%) (pyridoxine), and B12 (18%), niacin (12%), folic acid (14%), magnesium (17%), iodine (13%), zinc (18%), choline (12%), soft gel multivitamin (9%), and other supplements (probiotics, *n*-3 fatty acid) (15%) (Figure 3).

Moreover, around half of the children (46.6%) were provided a single vitamin/or mineral, 33.2% were using multi-vitamin-mineral supplements, while 20.2% were using both (single vitamin/or mineral and multi-vitamin-mineral). Almost all mothers (95.7%) reported supplementing their children’s diet based on a physician’s prescription. When asked about the reasons that led mothers to supplement their children’s diet, the majority of mothers (59.8%) admitted that it is to keep the child healthy, 30.5% stated that food nutrients are not sufficient to meet the child’s dietary needs (mostly among boy children (36.3% vs. girls (25.1%)), *p =* 0.04), and 9.7% reported doing so because their children have health-related problems that necessitate oral dietary supplementation (Appendix A).

### 3.6. The Frequency and Duration of Use of Single Vitamin/Or Mineral and Multi-Vitamin-Mineral Supplements among under-5 Children

Most vitamin and mineral supplements were provided for children once per day, including vitamin A (22.2%), vitamin E (16.4%), vitamin C (27.6%), vitamin B1 (15.2%), vitamin B2 (15.5%), vitamin B6 (14.9%), and vitamin B12 (17.9%). Additionally, niacin (12.2%), folic acid (13.7%), calcium (21.7%), magnesium (17.0%), iodine (12.5%), iron (58.9%), zinc (17.6%), and choline (12.1%) supplements were provided for the child mostly once per day. The same findings were obtained regarding the use of multivitamin drops, soft gel multivitamins, and other supplement products (probiotics, *n*-3 fatty acids), as they were consumed by most children once per day (60.2%, 9.1%, and 14.2%, respectively). Regarding the duration of use, almost all DSs were used for less than 3 months, including vitamin A (9.9%), vitamin E (7.6%), vitamin C (10.3%), vitamin B1 (7.9%), vitamin B2 (6.1%), vitamin B12 (6.4%), niacin (4.9%), folic acid (6.1%), calcium (9.1%), magnesium (6.8%), iodine (4.6%), choline (4.8%), multivitamin drops (41.6%), soft gel multivitamin (3.3%), and other supplement products (probiotics, *n*-3 fatty acids) (4.8%). However, some children used vitamin B6 (6.1%), iron (28.6%), and zinc (7.9%) for 3–6 months. Iron supplements were the most commonly used for a duration of more than one year (4.2%), followed by zinc (3.9%) and vitamin B12 (3.6%) (Table 4).

### 3.7. Mothers’ Sources of Information about Oral Dietary Supplementation

Mothers were asked to report the sources of information they frequently rely on to acquire information on DSs because it is one of the most important factors determining oral dietary supplementation practices among mothers and their under-5 children. Study findings showed that physicians were the primary information source for 63.6% of mothers, followed by online sources (16%) and pharmacists (12.3%). Moreover, the remaining mothers reported referring to other healthcare providers (3.2%), family members (1.6%), personal trainers/coaches (1.6%), friends (1.5%), and scientific books (1.1%) (Figure 4).

### 3.8. Side Effects Reported by Mothers Due to DSs Use

When mothers were asked about the adverse effects experienced after using DSs, a minority (3%) reported that DSs use has caused chronic constipation, and nausea, and 2.5% reported stomach pain. Other reported side effects were tachycardia (1.7%), headache (1.7%), skin itching (0.8%), and dizziness and confusion (0.4%) (Appendix A). Study findings also show that the use of DSs was predominant among male children compared to girls (68.3% vs. 61.2%, *p =* 0.01).

### 3.9. The Associations between Study Variables and Oral Dietary Supplementation among Mothers and Their under-5 Children

The associations between study variables and oral dietary supplementation among mothers and their under-5 children are shown in Table 5. Almost all mothers who supplemented their diets (90.2%) and their children’s diets (87.9%) were adults between the ages of 25 and 49; however, the mother’s age was not a significant correlate for DS use either among them (*p =* 0.28) or their under-5 children (*p =* 0.56). Regarding mothers’ weight status, about half of the mothers who were using DSs for themselves (54.7%) and their children (50.0%) had a normal body weight; although these findings showed no statistical significance (*p =* 0.60 and *p =* 0.50, respectively). Mothers using DSs for themselves and their under-5 children reported the same child’s birth weight (in grams) compared to their counterparts who did not supplement their diets (3177.0 ± 658 vs. 3160 ± 572, *p =* 0.61) and that of their children (3222.0 ± 592 vs. 3140.0 ± 631, *p =* 0.05). Furthermore, the highest use of DSs was among mothers (59.0%) and children (56.7%) who were residents in the Beirut and Mount Lebanon districts (*p* = 0.12 and *p =* 0.75, respectively). Almost all mothers who were supplementing their diets (99.9%, *p =* 0.179) and their children’s diets (99.7%, *p =* 0.670) were married. DS use among mothers and children was the highest in households composed of only one child (49.2% and 50.2%, respectively) and with no twins or triplets (96.0% and 94.4%, respectively); however, the number of children per household had a significant association with DS use only among mothers (*p =* 0.04). Even though DS use was significantly the highest among mothers of children aged 1–3 years old (35.1%), more than half of the mothers having children aged 0–6 months (55.0%) were using DSs, *p =* 0.01. The child’s gender did not correlate with the use of DSs among mothers (*p =* 0.98); however, DS use was significantly more prevalent among female children (52.2%) compared to boys (47.8%), *p =* 0.01 (Table 5).

When mothers reported ever breastfeeding their child, they reported the highest oral dietary supplementation practices for themselves (97.1%) and their children (96.6%); however, these findings showed no statistical significance (*p =* 0.06 and *p =* 0.05, respectively). Similarly, DS use was predominant among exclusively breastfed children (59.2%) compared to their counterparts (40.8%), *p =* 0.87. The use of DSs was higher among children who were introduced to their first solid food at an age of less than 6 months (54.8%), compared to 6 months and older (45.2%), *p =* 0.08. Despite the finding that about a third of mothers using DSs for themselves (75.4%) and their children (74.3%) had no experiences with COVID-19 infection, the latter did not show significant associations with mothers’ use of DSs and that of their children (*p =* 0.75 and *p =* 0.21, respectively). In the same manner, COVID-19 infection among children had no significant associations with their DS use and that of their mothers, with the majority of mothers using DSs for themselves (83.3%) and their children (81.8%) reported that their children had no experiences with the infection (*p =* 0.88 and *p =* 0.14, respectively). Non-healthcare working mothers reported higher supplementation practices for themselves (79.4%, *p =* 0.094) and their children (77.7%, *p =* 0.14) compared to their counterparts (healthcare workers). Although the DS use was more prevalent among non-pregnant women (vs. pregnant; 83.0% vs. 17%), 77.3% of pregnant women were using DSs, *p <* 0.001 (Table 5).

Most mothers who reported supplementing their diets and that of their children believed that DS use is important to maintain good health (68.6% and 76.5%; *p =* 0.01 and *p <* 0.001, respectively) and fight infections (66.4% and 72.5%; *p =* 0.001 and *p <* 0.001, respectively). Further, mothers denying the fact that the consumption of DSs could cause adverse effects had the highest prevalence of oral dietary supplementation (74.9%) with a higher tendency to supplement their children’s diets too (74.4%), with no statistical significance (*p =* 0.91 and *p =* 0.85, respectively). Interestingly, DSs use was predominant among mothers (57.5%) not perceiving that food nutrients are sufficient to meet dietary requirements, with a higher tendency to supplement their child’s diet too (54.0%) (*p =* 0.88 and *p =* 0.06, respectively). In addition, a higher proportion of mothers consenting that HPs could cause adverse effects when consumed with drugs reported the use of DSs than their counterparts (60.4% vs. 43.0% *p =* 0.001). Additionally, mothers refusing the claim that HPs are safe because they are from natural resources reported higher supplementation practices for themselves (61.6% vs. 40.5%, *p <* 0.001) and their children (71.2% vs. 61.0%, *p =* 0.03) than their counterparts consenting the claim. Notably, more than half of DS users (56.7%) were children of mothers who reported supplementing their diets too, *p <* 0.001 (Table 5).

### 3.10. Correlates of Dietary Supplementation among Mothers and Their under-5 Children in This Study: Regression Analysis

We applied the binary backward stepwise regression to point out the extent of contribution of the significant study variables to the use of DSs among mothers (model 1) and their under-5 children (model 2). Model 1 reveals that mothers having only one child were 70% more likely to use DSs than mothers of 3 children and more (OR = 0.3, CI = 0.1–0.9, *p =* 0.02). Moreover, mothers of children aged 0–6 months had a 60% higher probability to use DSs than mothers of 1–3 years old children (OR = 0.4, CI = 0.3–0.7, *p =* 0.002). Of interest, pregnancy increased the likelihood of using DSs among mothers by 7 times (OR = 7.0, CI = 3.2–15.2, *p <* 0.001). Mothers who believed that DSs use is necessary to maintain good health were 40% more likely to use DSs than their counterparts disagreeing with this claim (OR = 0.6, CI = 0.4–0.9, *p =* 0.01). Mothers who perceived the tendency of HPs to cause adverse effects when combined with drugs had about 2 times more possibility to use DSs than their counterparts (OR = 1.9, CI = 1.2–3.0, *p =* 0.01). In addition, mothers who disagreed that HPs are safe because they are from natural sources had 60% higher probability to use DSs than those who consented to the claim (OR = 0.4, CI = 0.3–0.6, *p <* 0.001) (Table 6).

As regards the DS use among under-5 children (model 2), Table 6 shows that the use of DSs among mothers had increased the probability of supplementing their child’s diet too by around 3 times (OR = 2.6, CI = 1.7–4.0, *p <* 0.001). Moreover, mothers consenting that DSs are necessary to maintain good health were 70% more likely to use DSs for their children than those who disagreed (OR = 0.3, CI = 0.2–0.6, *p <* 0.001) with this claim. Mothers perceiving the beneficial effect of the DSs to fight infections had a 50% (vs. mothers who refused this claim; OR = 0.5, CI = 0.3–0.8, *p =* 0.01) higher likelihood to supplement their child’s diet. (Table 6)

## 4. Discussion

To our knowledge, nationally and worldwide, this study is the first to evaluate maternal knowledge and attitudes towards DSs, along with the DS-related practices of mothers and their under-5 children and their correlates. Mothers presently explored knowledge deficits and unfavorable attitudes regarding the use of DSs. Among the mothers in this study, 47% were DS users, with the majority using vitamin D (82%). Moreover, 64.0% of mothers were providing DSs for their under-5 children, with multivitamin drops being the most commonly used (61.0%). “To keep the child healthy” was the frequently reported reason by mothers (60.0%) to supplement their child’s diet. Physicians were the primary information source about DSs for 63.6% of mothers. The usage of DSs among mothers was determined by their pregnancy status, child’s age, number of children per household, and their awareness and attitudes towards DSs. DS usage among under-5 children was correlated with maternal DSs use and mothers’ attitudes towards DSs. Mothers were 7 times more likely to use DSs if they were pregnant. The use of DSs by mothers increased the likelihood of supplementing their child’s diet by about threefold.

### 4.1. Mothers’ Knowledge and Attitudes towards Oral Dietary Supplementation

In the present study, mothers appeared to lack knowledge and appropriate attitudes toward oral dietary supplementation, with many expressing unfavorable beliefs that DSs are necessary to maintain good health, fight infections, and meet dietary needs. Additionally, a significant proportion (74.6%) of mothers did not perceive the potentiality of DSs in posing detrimental health effects. These findings are consistent with that observed among Chinese mothers of under-5 children who consented that some herbal supplements are beneficial to the immune system or to bring an improvement in health or well-being and that DSs might be used as medicines to prevent coughs or colds among their children [15]. In the Lebanese community, the oral dietary supplementation knowledge gap is a typical occurrence. Previous research efforts had explored the knowledge deficits related to the use of DSs among the general adult Lebanese population [2] and specific demographics, particularly Lebanese athletes [16]. However, the authors of the current study believe that the current findings are the most crucial since mothers are the primary health caretakers for their children, and a lack of awareness could cause them to start giving their children needless supplements. The irresponsible use of DSs by mothers could harm the child’s health for a variety of reasons. The American FDA does not perform rigorous testing and approval for DSs [17], which causes a lot of marketed supplement products to be of unknown safety and effectiveness and to be mislabeled. Children are the focus of particular attention because there has not been a sufficient number of trials on the efficacy of DS in this age group [18]. Even though the American Academy of Family Physicians (AAFP) and the AAP have endorsed recommendations for DS use among children, the literature on the clinical effectiveness and safety of herbal products and DSs in children and infants is still scant [18]. In a study on the market analysis of vitamin supplementation in infants and children, it was shown that vitamin D was the only vitamin present in the supplements containing at or below the Recommended Dietary Allowance (RDA) value for pediatrics [19]. The remaining vitamins presented in the Institute of Medicine (IOM) list contained values above the RDA values; for example, vitamin levels varied from a low of 13% of adequate intake for choline to a high of 936% of adequate intake for biotin [19].

Considering these findings, mothers should be the focus of education and counseling sessions provided by healthcare providers and specialists to help them answer these questions correctly: “when?”, “how much?”, “which?”, and “under what circumstances?” she could supplement her child’s diet. Nevertheless, the blame is beyond the mother’s knowledge in the presence of aggressive marketing approaches for the use of DSs in the pediatric population. An examination of marketing techniques used to promote children’s vitamins in parenting magazines showed that from a total of 135 magazines reviewed across four years, there were 207 advertisements for children’s vitamins, all in the form of a chewy or gummy [20]. What is worth mentioning is that none of these advertisements included a dosage or a warning regarding the DSs [20]. Moreover, almost all (92.3%) included a cartoon in the advertisement, which is a tactic to catch the child’s attention [20]. Therefore, healthcare providers should be diligent about discussing the appropriateness of the use of DSs with the caregivers and identify the best strategies to improve the nutritional quality of the child’s diet without encouraging unnecessary dietary supplementation.

### 4.2. Practices Related to Oral Dietary Supplementation among Mothers

The present study indicates that about half of the mothers (47%) reported using one or more supplement products, with a predominant use of vitamin D (82%), followed by iron (67%) and calcium (61%) supplements. The latter finding comes hand-in-hand with the recently observed findings among American mothers, with 45% of them reported to use one or more supplement products [21]. Our finding may be partially explained by the fact that our sample population included mothers who were currently breastfeeding their babies. Breastfeeding mothers have higher nutrient requirements, which necessitate the use of DSs. A recent systematic review [22] explored that maternal dietary vitamin and/or mineral supplementation, particularly fat-soluble vitamins, vitamins B1, B2, and C might be reflected in the breast milk composition. Moreover, the Dietary Guidelines for Americans recommend that lactating parents consume 290 mcg of iodine and 550 mg of choline daily throughout the first year postpartum [23], and thus, this might be one possible explanation of our finding that 16% and 13% of mothers in this study reported the use of iodine and choline supplements, respectively. Further, about 10% of mothers in the present study were pregnant at the time of collecting data. Multiple studies worldwide have found that pregnant women’s intake of micronutrients, such as folic acid, iron, and vitamin D, is below the recommendation [24]. Nutritional deficiencies during pregnancy can have serious health consequences, such as anemia in pregnant women, and neural tube defects (NTDs) in fetuses [25]. Moreover, antenatal dietary supplementation resulted in lower stunting prevalence until 5 years of age in trials performed in Indonesia [26] and Bangladesh [27]. Even though the beneficial effect of dietary supplementation in the preconception period and during conception has been recorded primarily for folic acid, women of childbearing age and pregnant women have increased the use of other supplements, including the omega-3 polyunsaturated fatty acid, calcium, vitamin D, zinc, multivitamins, and herbal ingredients such as Echinacea and ginger [28]. Dietary supplementation has become a common practice among pregnant women to meet dietary needs and avoid conception complications. Nonetheless, there is little evidence to support vitamin and mineral supplements during pregnancy unless nutrition inadequacy is suspected. Moreover, it is crucial to consider that this study was carried out when the COVID-19 pandemic was severely affecting the country and there was a strict national lockdown. People in the country relied on complementary protective approaches such as DSs and herb-based products with a belief to have defensive benefits against coronavirus, as evident by data of a recent study that assessed the usage, knowledge, and attitudes towards dietary supplementation before and during the COVID-19 pandemic among Lebanese people [2]. Thus, our findings are warranted.

### 4.3. Practices Related to Oral Dietary Supplementation among under-5 Children

The studies evaluating DSs usage among children under the age of five are currently scant worldwide, making it difficult to compare our findings to that of other demographics, particularly given the lack of published data regarding DSs usage among under-5 children in the Arab world. However, our observed prevalence of oral dietary supplementation (64.4%) is higher than that reported among under-5 children in Germany (42%) [29], the U.S. ((18.2%) [30]; (8–31%) [31]), Australia (22.6%) [15], and China (32.4% [15]; 34.9% [32]). To add value to the discussion of our findings, we also compared our obtained prevalence to data reported among children belonging to other age groups. Our prevalence rate exceeded that reported in Japan among 6–12-year-old children (6.8%) [33], and school-aged children (21.6%) [34]. It also appears to exceed the rates observed in the U.S. among less than 19 years old children (31%) [35]. Consistent data are also observed when comparing our prevalence to that observed in Poland among (under 12 years old (39.5%) [36]; 7–14-year-old children (33.8%) [37]), China (22%; 6–12-year-old children) [38], and Korea (20.3%; 1–18 years old) [39]. On the contrary, 71.4% of mothers from south-eastern Poland admitted to supplying DSs to their children (from birth up to 18 years old), surpassing our obtained prevalence by nearly 7% [40]. Moreover, an earlier study also reported just near the prevalence of DSs use (52%) among children of kindergarten age (4–6 years old) in Hong Kong, China [41]. All in all, in this study, Lebanese children showed higher projections towards using DSs than under-5 children from other countries and even children older than 5 years old. However, as this investigation was carried out during the COVID-19 pandemic period, our findings are justifiable. Children were mistakenly assumed by their parents to have additional risks of catching the coronavirus and having more severe complications than other age groups. Moreover, it is important to keep in mind that under-5 children, particularly breastfed, might be in need of oral dietary supplementation more than older children.

Our study findings also explored that among children using DSs, multivitamin drops were most used by 61%. The latter findings are in concordance with those observed among Australian [15], Chinese [32], and American under-5 children [30,31] where the use of multi-vitamin-mineral supplements was predominant. In the light of this, the AAP recommends that a child might use vitamin and mineral supplements only after consultation with a pediatrician or healthcare provider [5]. According to a national investigation in the U.S. on 1251 parents with at least one child aged 1–10, about a third of parents reported that their child is a picky eater, and a third believe that the child is not eating enough fruits and vegetables, and 30% worry that children are not getting enough vitamins and minerals [42]. Mothers usually assume that their children are not getting enough vitamins and minerals from their diets; in fact, breakfast cereal, milk, and orange juice, all of which are frequently fortified with B vitamins, vitamin D, calcium, and iron might be providing more nutrients than mothers realize [3]. A multivitamin supplement might be helpful for a child if he/she has a delay in physical and developmental growth (failure to thrive), has certain chronic diseases or food allergies, and follows a restrictive diet, such as a strict vegan diet; otherwise, it has no additional benefits on the child’s health [5]. Moreover, the AAP recommends that exclusively breastfed infants be given iron supplements, starting at 4 months of age, when their innate iron stores start to deplete [43]. In addition, the NHS recommends that breastfed infants aged 6 months to 5 years must be given vitamins A, C, and D supplements every day; nonetheless, this is not necessary for formula-fed infants [44]. Thus, one possible explanation for observing the findings that iron, vitamin C, and vitamin A are supplemented for 60.0%, 28.0%, and 23.0% of children in the current study, respectively, is the promising exclusive breastfeeding prevalence rate (59%) recorded among this sampled population of mother–children dyads. Also noteworthy is the finding that almost all mothers (95.7%) were providing their children with DSs because of physician prescription, supporting the finding observed in Germany [29]; thus, our latter finding suggests that mothers in the present study are most probably and hopefully not practicing unnecessary dietary supplementation practices for their children. However, mothers’ misreporting is a predicted bias of this study, and we could not guarantee that mothers were guided by pediatricians about supplementing their child’s diet. Among the reasons mothers reported supplementing their child’s diet, keeping the child healthy (60%), food nutrients are not enough to meet the child’s dietary needs (30.6%), and that the child has a medical condition that necessitates the use of DSs (9.4%). The latter findings are similar to the reported reasons by mothers in the U.S. which were as follows: “maintain health”, “supplement the diet, food not enough”, and “improve overall health” [30]. Therefore, mothers should be referred to dietitians and public health nutritionist where they can perform a detailed nutritional assessment and provide guidance about the adequacy of their children’s diets.

### 4.4. The Correlates of Oral Dietary Supplementation among Mothers and Their under-5 Children

In our study, mothers were 7 times more likely to use DSs when pregnant. This finding is warranted, as many clinicians recommend DS use (such as iron, folic acid, and iodine) during pregnancy which is a critical period of the life cycle when nutrient requirements of women increase. Again, the use of DSs by pregnant women should be practiced with caution, and this depends on demographic, sociologic, and economic factors. Another finding of our study is that mothers who reported using DSs were more likely to provide their children with DSs by around 3 times. According to Centers for Disease Control and Prevention (CDC) report [45], children with parents who use DSs or other complementary health approaches are more likely to use them than other children whose parents do not. Additionally, supporting our latter finding, a previous cross-sectional observation showed that 85% of the supplements used by American children are based on either self-selection or decisions made by parents, who are most probably not qualified to guide the frequency and the dose of the supplementation in their children’s diet [36]. Mothers of 0–6-month-old children were 60% more likely to supplement their diets than others having children aged 1–3 years old in our study. This may be because mothers of infants aged 0–6 months are more likely to nurse their children, increasing their demand for using DSs. Our study also showed that mothers of one child were 70% more likely to use DSs than mothers of 3 children and more. We could relate this finding to the fact that households with many children have additional financial needs that may reduce the sale of DS products, as many of them are considered expensive in the Lebanese market. In addition, mothers believing that DSs are necessary to maintain good health were more likely to supplement their diets and their children’s diet in this study. As well, children of mothers perceiving the tendency of DSs to fight infections had a higher probability to be provided with DSs. It is well known that knowledge, attitudes, and behaviors of any subject are interrelated. As a result, mothers’ perceptions about oral dietary supplementation lead them to use DSs for their children, which may or may not be necessary; otherwise, it may pose health risks that outweigh any other benefits.

### 4.5. Reported Side Effects Due to DSs Use

Our study showed that among mothers who reported experiencing side effects after DS use, chronic constipation, nausea, and stomach pain were the most frequent. Iron supplements might cause chronic constipation in certain circumstances, especially among pregnant women, as pregnancy causes changes in bowel movements due to hormonal changes [46]. Moreover, calcium, vitamin C, and zinc supplements have been linked causing nausea and stomachache in many people, especially when consumed in toxic doses [46].

At last, DSs, which include vitamins and minerals, are products made to complement the daily consumption of nutrients. While many are risk-free and have considerable positive health effects, some can also be harmful, particularly if used excessively.

### 4.6. Limitations and Strengths of the Study

The findings of this study should be treated with caution, as some limitations are inevitable. First, it is not possible to reach causal inferences as this study has a cross-sectional design. Second, misreporting is possible with a particular concern of having recall bias, as mothers were asked to record all supplement products used by herself and her child currently or before collecting data. Third, conducting such an investigation in a period when the COVID-19 pandemic was emerging and tragically spreading in the country might lead to non-typical estimation of oral dietary supplementation among mothers and their under-5 children. However, the main strength of this study is that, as far as we know, it is the first of the region’s kind to assess the oral dietary supplementation patterns of a representative sample of mother–under-5 child dyads in Lebanon.

## 5. Conclusions

In closing, this study shows that oral dietary supplementation among a representative sample of Lebanese mother–under-5 child dyads is common, given that many mothers lack awareness and hold unfavorable views regarding the use of DSs. Most mothers reported supplementing their children’s diet to keep them healthy. Moreover, mothers’ use of DSs increased seven times during pregnancy, and maternal DS use increased the likelihood that the child’s diet would be supplemented by around three times.

### Future Directions

Whenever possible, it is preferable that infants and young children meet their dietary needs through a well-balanced and diversified diet rather than oral DSs. There is an increase in advertisements that promote supplements products as preventives or cures for a variety of childhood health conditions, leading parents to fall for the products and disregard proven and perhaps essential treatment approaches for their children. Before initiating dietary supplementation, consultation with healthcare professionals is recommended. Moreover, a “call to action” for the healthcare community to educate parents about DSs and enhance their knowledge that a balanced diet and a healthy lifestyle are the best sources of vitamins, especially during growth. The authors motivate future research efforts to examine total nutrient intake from food, beverages, and DSs among Lebanese mothers and their under-5 children and evaluate the nutrient adequacy of their diets.

## Figures and Tables

**Figure 1 foods-11-03005-f001:**
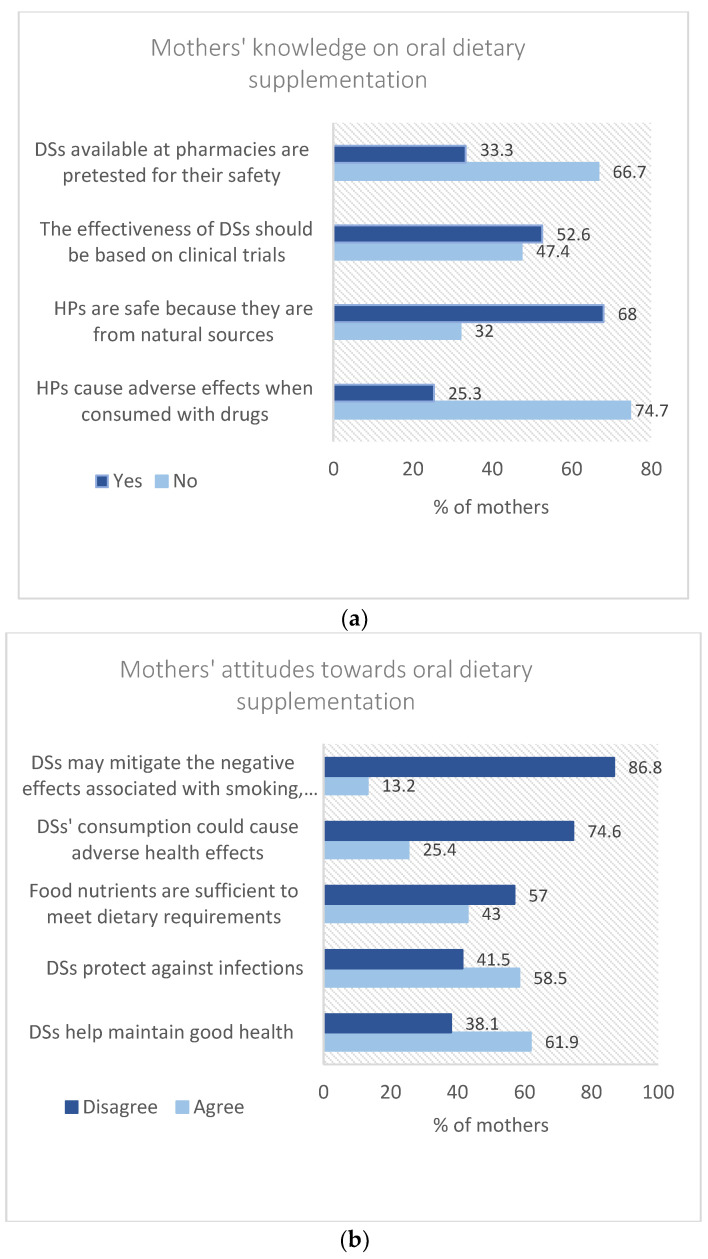
(**a**) Mothers’ knowledge on oral dietary supplementation. HPs: herbal products; DSs: dietary supplements. (**b**) Mothers’ attitudes towards oral dietary supplementation. DSs: dietary supplements. The rating scale for mothers’ attitudes towards oral dietary supplementation was: Strongly agree, Agree, Neutral, Disagree, and Strongly Disagree. In order to facilitate to data presentation and analysis, we assumed that a neutral response implies disagreement, so we merged neutral with the disagree and strongly disagree options.

**Figure 2 foods-11-03005-f002:**
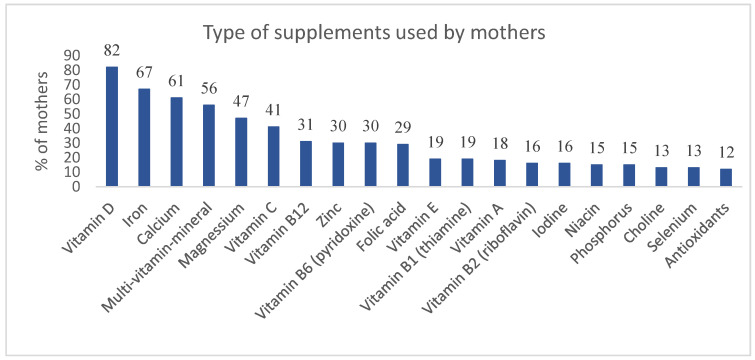
Type of supplements used by mothers.

**Figure 3 foods-11-03005-f003:**
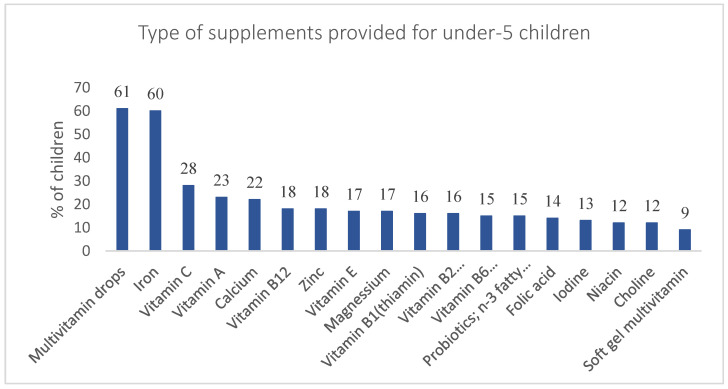
Types of supplements provided for under-5 children.

**Figure 4 foods-11-03005-f004:**
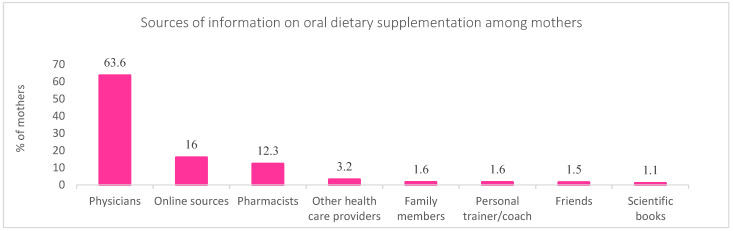
Mothers’ sources of information on oral dietary supplementation.

**Table 1 foods-11-03005-t001:** General characteristics of the mothers in this study.

	Mothers (*n* = 511)
	Mean ± SD
Age (years)	30.25 ± 4.98
BMI (kg/m^2^)	24.85 ± 4.53
	*n* (%)
Age category	
Youth (18–24 years old)	58 (11.4)
Adults (25–49 years old)	453 (88.6)
Weight status (*n* = 503)	
Underweight	21 (4.2)
Normal weight	262 (52.0)
Overweight	153 (30.4)
Obese	67 (13.4)
Current Residence	
Beirut and Mount Lebanon	288 (56.4)
North Lebanon and Akkar	124 (24.2)
Beqaa and Baalbeck/Hermel	39 (7.6)
South Lebanon and Nabatieh	60 (11.8)
Marital status	
Married	509 (99.6)
Divorced	2 (0.4)
Number of children per household	
One child	241 (47.2)
2–3 children	240 (46.6)
More than 3 children	30 (6.2)
Mothers of twins or triplets	
Yes	26 (5.0)
No	485 (95.0)
The mother is currently pregnant	
Yes	53 (10.3)
No	458 (89.7)
Family income per month (*n* = 510)	
Less than LBP 750,000	44 (8.6)
LBP 750,000–2,250,000	304 (59.6)
More than LBP 2,250,000	162 (31.8)
The mother is currently working	
Yes	219 (42.8)
No	292 (57.2)
Healthcare specialization among working mothers (*n* = 219)	
Yes	56 (25.6)
No	163 (74.4)
COVID-19 infection	
Yes	123 (24.1)
No	388 (75.9)
Household crowding index ^1^ (Mean ± SD)	1.03 ± 0.40

^1^ Number of co-residents (excluding newborns) divided by number of rooms (excluding kitchens and bathrooms).

**Table 2 foods-11-03005-t002:** General characteristics of under-5 children in this study.

	Overall (*n* = 511)	Boys (*n* = 230)	Girls (*n* = 281)	
	Mean ± SD	Mean ± SD	Mean ± SD	*p*-Value
Age (months)	18.7 ± 15.5	18.0 ± 15.5	19.5 ± 15.6	0.31
Weight at birth in grams	3168.9 ± 617.9	3074.18 ± 622.2	3283.35 ± 594.2	<0.001 *
Height at birth in cm	49.5 ± 5.2	49.2 ± 4.74	49.8 ± 5.7	0.31
	*n* (%)	*n* (%)	*n* (%)	
Age categories				0.13
0–6 months	124 (25.3)	54 (24.4)	70 (25.7)	
6–12 months	105 (21.3)	38 (17.3)	67 (24.8)	
1–3 years	179 (36.4)	91 (40.9)	88 (32.5)	
3–5 years	84 (17.0)	38 (17.4)	46 (17.0)	
COVID-19 infection				0.13
Yes	83 (16.4)	32 (13.7)	52 (18.6)	
No	428 (83.6)	198 (86.3)	229 (81.4)	
The child is ever breastfed				0.26
No	25 (4.8)	14 (5.9)	11 (3.9)	
Yes	486 (95.2)	216 (94.1)	270 (96.1)	
The child is exclusively breastfed (*n* = 486)				0.24
Less than 6 months of age	287 (59.1)	121 (56.1)	166 (61.3)	
6 months and above	199 (40.9)	95 (43.9)	105 (38.7)	
Initiation of complementary feeding (*n* = 373)				0.001 *
Less than 6 months of age	193 (51.7)	98 (61.5)	95 (44.5)	
6 months and above	180 (48.2)	61 (38.5)	118 (55.5)	

* Significant at *p*-value < 0.05 for χ 2 test.

**Table 3 foods-11-03005-t003:** The frequency and duration of use of single vitamin/or mineral and multi-vitamin-mineral supplements among mothers.

	MVM * *n* (%)	Antioxidants *n* (%)	Vitamin A *n* (%)	Vitamin C *n* (%)	Vitamin D *n* (%)	Vitamin E *n* (%)	Vitamin B1 *n* (%)	Vitamin B2 *n* (%)	Vitamin B6 *n* (%)	Vitamin B12 *n* (%)
Daily	95 (39.2)	15 (6.2)	28 (11.6)	49 (20.3)	73 (30.2)	29 (12.0)	26 (10.7)	23 (9.5)	39 (16.1)	37 (15.3)
2–6 times per week	24 (9.9)	6 (2.4)	9 (3.7)	30 (12.4)	66 (27.3)	8 (3.3)	5 (2.1)	4 (1.6)	22 (9.1)	22 (9.1)
Once per week	13 (5.4)	6 (2.4)	5 (2.0)	16 (6.6)	35 (14.5)	7 (2.9)	11 (4.5)	10 (4.1)	11 (4.5)	10 (4.1)
Once per month	3 (1.2)	1 (0.3)	2 (0.8)	3 (1.2)	24 (10.0)	2 (0.8)	4 (1.6)	3 (1.2)	1 (0.3)	5 (2.1)
	Niacin *n* (%)	Folic acid *n* (%)	Calcium *n* (%)	Magnesium *n* (%)	Iodine *n* (%)	Iron *n* (%)	Zinc *n* (%)	Choline *n* (%)	Phosphorus *n* (%)	Selenium *n* (%)
Daily	22 (9.1)	47 (19.4)	95 (39.3)	71 (29.4)	22 (9.1)	122 (50.4)	35 (14.5)	17 (7.0)	19 (7.9)	20 (8.3)
2–6 times per week	7 (2.9)	14 (5.8)	34 (14.0)	32 (13.2)	6 (2.4)	24 (10.0)	20 (8.3)	2 (0.8)	8 (3.3)	2 (0.8)
Once per week	5 (2.1)	7 (2.9)	8 (3.3)	10 (4.1)	10 (4.1)	13 (5.4)	16 (6.6)	10 (4.1)	6 (2.5)	8 (3.3)
Once per month	1 (0.3)	2 (0.8)	9 (3.7)	1 (0.3)	1 (0.3)	5 (2.1)	1 (0.3)	2 (0.8)	2 (0.8)	1 (0.3)

* MVM: Multi-vitamin-mineral supplements.

**Table 4 foods-11-03005-t004:** The frequency and the duration of use of single vitamin/or mineral and multi-vitamin-mineral supplements among under-5 children.

Among Supplement Users (*n* = 328)	Vitamin A	Vitamin E	Vitamin C	Vitamin B1	Vitamin B2	Vitamin B6	Vitamin B12	Niacin	Folic Acid
Frequency of use	*n* (%)	*n* (%)	*n* (%)	*n* (%)	*n* (%)	*n* (%)	*n* (%)	*n* (%)	*n* (%)
Once per day	73 (22.2)	54 (16.4)	91 (27.6)	50 (15.2)	51 (15.5)	49 (14.9)	59 (17.9)	40 (12.2)	45 (13.7)
Twice per day	2 (0.6)	1 (0.3)	2 (0.6)	1 (0.3)	0 (0)	0 (0)	0 (0)	0 (0)	0 (0)
**Duration of use**	***n*** (**%**)	***n*** (**%**)	***n*** (**%**)	***n*** (**%**)	***n*** (**%**)	***n*** (**%**)	***n*** (**%**)	***n*** (**%**)	***n*** (**%**)
Less than 3 months	33 (9.9)	25 (7.6)	34 (10.3)	26 (7.9)	20 (6.1)	13 (4.0)	21 (6.4)	16 (4.9)	20 (6.1)
3–6 months	22 (6.8)	16 (4.9)	30 (9.1)	14 (4.3)	17 (5.2)	20 (6.1)	19 (5.8)	10 (3.1)	10 (3.1)
1 year	13 (3.8)	8 (2.4)	19 (5.8)	5 (1.5)	7 (2.1)	9 (2.7)	7 (2.1)	7 (2.1)	7 (2.1)
More than 1 year	8 (2.3)	6 (1.8)	10 (3.1)	6 (1.8)	6 (1.8)	6 (1.8)	12 (3.6)	6 (1.8)	8 (2.3)
	**Calcium**	**Magnesium**	**Iodine**	**Iron**	**Zinc**	**Choline**	**Multivitamin drops**	**Soft gel multivitamin**	**Probiotics; *n*-3 fatty acid**
**Frequency of use**	***n*** (**%**)	***n*** (**%**)	***n*** (**%**)	***n*** (**%**)	***n*** (**%**)	***n*** (**%**)	***n*** (**%**)	***n*** (**%**)	***n*** (**%**)
Once per day	71 (21.7)	56 (17.0)	41 (12.5)	194 (58.9)	58 (17.6)	40 (12.1)	198 (60.2)	30 (9.1)	47 (14.2)
Twice per day	1 (0.3)	0 (0)	0 (0)	3 (1.0)	0 (0)	0 (0)	1 (0.3)	0 (0)	3 (0.9)
**Duration of use**	***n*** (**%**)	***n*** (**%**)	***n*** (**%**)	***n*** (**%**)	***n*** (**%**)	***n*** (**%**)	***n*** (**%**)	***n*** (**%**)	***n*** (**%**)
Less than 3 months	30 (9.1)	20 (6.8)	15 (4.6)	51 (15.5)	16 (4.8)	16 (4.8)	137 (41.6)	11 (3.3)	16 (4.8)
3–6 months	25 (7.6)	21 (6.4)	11 (3.3)	94 (28.6)	19 (7.9)	10 (3.0)	43 (13.0)	10 (3.0)	8 (2.4)
1 year	6 (1.8)	6 (1.8)	7 (2.1)	39 (11.8)	10 (3.0)	9 (2.7)	13 (3.9)	5 (1.5)	16 (4.8)
More than 1 year	11 (3.3)	8 (2.4)	8 (2.4)	14 (4.2)	13 (3.9)	6 (1.8)	6 (1.8)	4 (1.2)	10 (3.0)

**Table 5 foods-11-03005-t005:** The associations between study variables and oral dietary supplementation among mothers and their under-5 children.

Variables	Dietary Supplementation among Mothers		Dietary Supplementation among under-5 Children		*p*-Value ^(a)^	*p*-Value ^(b)^
	No (*n =* 269)	Yes (*n* = 242)	No (*n* = 182)	Yes (*n* = 329)		
	*n* (%)	*n* (%)	*n* (%)	*n* (%)		
Mother’s age					0.28	0.56
Youth (18–24 years old)	35 (12.9)	23 (9.8)	19 (10.2)	39 (12.1)		
Adults (24–49 years old)	234 (87.1)	219 (90.2)	163 (89.8)	290 (87.9)		
Mother’s weight status					0.60	0.50
Underweight	11 (4.2)	10 (4.2)	5 (2.8)	16 (5.0)		
Normal weight	132 (49.6)	130 (54.7)	100 (55.6)	162 (50.0)		
Overweight	88 (33.1)	65 (27.4)	53 (29.3)	100 (31.1)		
Obese	35 (13.1)	32 (13.7)	22 (12.3)	45 (14.0)		
Child’s birth weight (mean ± SD)	3177.0 ± 658	3160 ± 572	3222.0 ± 592	3140.0 ± 631	0.61	0.05
Residency					0.12	0.75
Mount Lebanon/Beirut	145 (54)	143 (59)	102 (55.8)	186 (56.7)		
North Lebanon and Akkar	61 (22.7)	63 (25.9)	42 (23.1)	82 (24.8)		
Beqaa and Baalbeck-Hermel	23 (8.6)	16 (6.5)	13 (7.3)	26 (7.8)		
South Lebanon and Nabatieh	40 (14.7)	20 (8.6)	25 (13.8)	35 (10.7)		
Marital Status					0.18	0.67
Married	267 (99.3)	242 (99.9)	181 (99.4)	328 (99.7)		
Divorced	2 (0.7)	0 (0.1)	1 (0.6)	1 (0.3)		
Number of children					0.04 *	0.34
One child	122 (45.5)	119 (49.2)	76 (41.9)	165 (50.2)		
2–3	124 (46.2)	116 (47.8)	91 (50.1)	149 (45.2)		
More than 3	22 (8.3)	7 (3.0)	14 (8.0)	15 (4.6)		
Mother of twins or triplets						
No	252 (94.0)	233 (96.0)	175 (96.0)	310 (94.4)	0.34	0.41
Yes	16 (6.0)	10 (4.0)	7 (4.0)	19 (5.6)		
Age of the child					0.01 *	0.22
0–6 months	57 (21.8)	69 (29.5)	45 (25.0)	82(25.7)		
6–12 months	51 (19.4)	62 (26.6)	35 (19.2)	78 (24.8)		
1–3 years	124 (47.8)	81 (35.1)	75 (42.6)	130 (41.4)		
3–5 years	28 (10.9)	20 (8.8)	23 (13.1)	25 (14.8)		
Gender of the child					0.98	0.01 *
Girl	148 (55.0)	133 (54.9)	109 (60.1)	172 (52.2)		
Boy	121 (45.0)	109 (45.1)	73 (39.9)	157 (47.8)		
Child is ever breastfed					0.06	0.05
No	17 (6.5)	7 (2.9)	13 (7.3)	11 (3.4)		
Yes	251 (93.5)	235 (97.1)	169 (92.7)	318 (96.6)		
Child is exclusively breastfed					0.21	0.87
Less than 6 months	155 (61.6)	132 (56.2)	99 (58.6)	188 (59.2)		
More than 6 months	96 (38.4)	103 (43.8)	70 (41.4)	129 (40.8)		
Initiation of complementary feeding					0.17	0.08
Less than 6 months	109 (55.0)	84 (48.1)	55 (45.4)	137 (54.8)		
6 months and more	89 (45.0)	91 (51.9)	67 (54.6)	113 (45.2)		
History of COVID-19 in mothers					0.75	0.21
No	205 (76.5)	183 (75.4)	144 (78.9)	244 (74.3)		
Yes	63 (23.5)	60 (24.6)	38 (21.1)	85 (25.7)		
History of COVID-19 in children					0.88	0.14
No	225 (83.9)	202 (83.3)	158 (86.9)	269 (81.8)		
Yes	43 (16.1)	40 (16.7)	23 (13.1)	60 (18.2)		
Mother as healthcare worker (*n* = 219 working mothers)					0.09	0.14
No	72 (69.1)	91 (79.4)	55 (68.8)	108 (77.7)		
Yes	32 (30.9)	24 (20.6)	25 (31.2)	31 (22.3)		
Mother is pregnant					<0.001 **	0.79
No	257 (95.7)	201 (83.0)	164 (90.1)	294 (89.4)		
Yes	12 (4.3)	41 (17.0)	18 (9.9)	35 (10.6)		
Mothers’ attitudes: DSs are necessary to maintain good health					0.01 *	<0.001 **
Agree	149 (55.9)	161 (68.6)	66 (36.1)	245 (76.5)		
Disagree	118 (44.1)	74 (31.4)	116 (63.9)	76 (23.5)		
Mothers’ attitudes: DSs help fight infections by strengthening the immunity					0.001 *	<0.001 **
Agree	137 (51.5)	157 (66.4)	62 (33.8)	233 (72.5)		
Disagree	129 (48.5)	80 (33.6)	120 (66.2)	88 (27.5)		
Mothers’ attitudes: DS use could cause health adverse effects					0.91	0.85
Agree	68 (25.7)	60 (25.1)	45 (25.0)	83 (25.6)		
Disagree	198 (74.3)	179 (74.9)	136 (75.0)	241 (74.4)		
Mothers’ attitudes: Food nutrients are sufficient to meet dietary requirements with no need for dietary supplementation					0.88	0.06
Agree	115 (43.3)	103 (42.5)	68 (37.4)	150 (46.0)		
Disagree	151 (56.7)	139 (57.5)	114 (62.6)	176 (54.0)		
Mothers’ knowledge: HPs can cause adverse effects if consumed with drugs					0.001 *	0.84
No	218 (81.0)	164 (67.7)	137 (75.4)	245 (74.4)		
Yes	51 (19.0)	78 (32.3)	45 (24.6)	84 (25.6)		
Mothers’ knowledge: HPs are safe because they are from natural sources					<0.001 **	0.03 *
No	63 (23.3)	101 (41.8)	47 (25.8)	116 (35.5)		
Yes	206 (76.7)	140 (58.2)	135 (74.2)	211 (64.5)		
Mothers’ practices: DS use among mothers					NA	<0.001 **
No	NA	NA	126 (69.4)	142 (43.3)		
Yes	NA	NA	56 (30.6)	187 (56.7)		

^(a)^*p*-value related to dietary supplementation among mothers. ^(b)^*p*-value related to dietary supplementation among under-5 children. * Significant at *p*-value < 0.05 for χ 2 test. ** Significant at *p*-value < 0.001 for χ 2 test.

**Table 6 foods-11-03005-t006:** Correlates of oral dietary supplementation among mothers and their under-5 children in this study: Regression analysis.

Model 1: Binary Backward Stepwise Regression Taking Oral Dietary Supplementation among Mothers as the Dependent Variable (Yes vs. No (Reference))		
	aOR (95% CI)	*p*-Value
Number of children (Reference: One child)		
2–3 children	1.1 (0.7–1.6)	0.97
More than 3	0.3 (0.1–0.9)	0.02 *
Age of the child (Reference: 0–6 months)		
6–12 months	1.2 (0.7–2.1)	0.56
1–3 years	0.4 (0.3–0.7)	0.002 *
3–5 years	0.6 (0.3–1.2)	0.16
Mother is pregnant (Reference: No)		
Yes	7.0 (3.2–15.2)	<0.001 **
Mothers’ attitude: DSs are necessary to maintain good health (Reference: Agree)		
Disagree	0.6 (0.4–0.9)	0.01 *
Mothers’ knowledge: HPs can cause adverse effects if consumed with drugs (Reference: No)		
Yes	1.9 (1.2–3.0)	0.01 *
Mothers’ knowledge: HPs are safe because they are from natural sources (Reference: No)		
Yes	0.4 (0.3–0.6)	<0.001 **
Model 2: Binary backward stepwise regression taking oral dietary supplementation among under-5 children as the dependent variable (Yes vs. No (reference))		
	AOR (95% CI)	*p*-value
DS use among mothers (Reference: No)		
Yes	2.6 (1.7–4.0)	<0.001 *
Mothers’ attitude: DSs are necessary to maintain good health (Reference: Agree)		
Disagree	0.3 (0.2–0.6)	<0.001
Mothers’ attitude: DSs help fight infections by strengthening the immunity (Reference: Agree)		
Disagree	0.5 (0.3–0.8)	0.01 *

aOR: Adjusted odds ratio; CI: confidence interval; * Significant at *p*-value < 0.05 for binary backward stepwise regression analytical test. ** Significant at *p*-value < 0.001 for binary backward stepwise regression analytical test.

## Data Availability

The original contributions presented in the study are included in the article/supplementary material, further inquiries can be directed to the corresponding author.

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
