# Peer review of "Prevalence of Essential Nutrient Supplement Use and Assessment of the Knowledge and Attitudes of Lebanese Mothers towards Dietary Supplement Practices in Maternal, Infancy and Preschool Ages: Findings of a National Representative Cross-Sectional Study"

_foods, 2022, doi:10.3390/foods11193005_

Round 1

Reviewer 1 Report

I have enclosed a word file in this review to consider. 

Reviewer 2 Report

This manuscript fulfills a gap in knowledge about use and parental knowledge about dietary supplements that can be useful to policy decision makers, and community health and healthcare professionals.

The authors have chosen to adapt a CDC questionnaire for their audience, although the nature of the adaptation (and translation?) is/are not fully explained.  It was not explained how these adaptations were tested to ensure that the original concepts were retained after the adaptation and translation?   In interpreting the findings are the authors considering herbal products and one example of dietary supplements or are you considering these as two different entities.  In your introduction you mention the US definition is this also the definition in general use in Lebanon? 

The article is strong on the summarizing the data collected and relationships within the data, but doesn’t include much about interpreting the findings into potential actions that could be taken to address the needs identified.  Should there be a section that summarizes the various issues or needs identified throughout the previous sections, lack of research and monitoring data, knowledge gaps, advertising for supplements, sources of information that may not have extensive knowledge, etc.   This section could also identify some potential solutions and who might be able to address the issues/needs.  Then this content could be reflected in the conclusion statement rather than just one statement that parents need to consult with healthcare provider prior to providing DS to their children?

Specific comments are attached:
